# Overlapping of Pulmonary Fibrosis of Postacute COVID-19 Syndrome and Tuberculosis in the Helminth Coinfection Setting in Sub-Saharan Africa

**DOI:** 10.3390/tropicalmed7080157

**Published:** 2022-07-30

**Authors:** Luis Fonte, Armando Acosta, María E. Sarmiento, Mohd Nor Norazmi, María Ginori, Yaxsier de Armas, Enrique J. Calderón

**Affiliations:** 1Department of Parasitology, Institute of Tropical Medicine “Pedro Kourí”, Havana 11400, Cuba; 2School of Health Sciences, Universiti Sains Malaysia, Kubang Kerian 16150, Kelantan, Malaysia; ducmar13@gmail.com (A.A.); mariesarmientogarcia@gmail.com (M.E.S.); norazmimn@usm.my (M.N.N.); 3Department of Teaching, Polyclinic “Plaza de la Revolución”, Havana 11300, Cuba; maginorig@infomed.sld.cu; 4Department of Clinical Microbiology Diagnostic, Hospital Center of Institute of Tropical Medicine “Pedro Kourí”, Havana 11400, Cuba; yaxsier2017@gmail.com; 5Department of Pathology, Hospital Center of Institute of Tropical Medicine “Pedro Kourí”, Havana 11400, Cuba; 6Instituto de Biomedicina de Sevilla, Hospital Universitario Virgen del Rocío, Consejo Superior de Investigaciones Científicas, Universidad de Sevilla, 41013 Sevilla, Spain; 7Centro de Investigación Biomédica en Red de Epidemiología y Salud Pública (CIBERESP), 28029 Madrid, Spain; 8Depatamento de Medicina, Facultad de Medicina, Universidad de Sevilla, 41009 Sevilla, Spain

**Keywords:** pulmonary fibrosis, postacute COVID-19 syndrome, tuberculosis, helminth coinfection, sub-Saharan Africa

## Abstract

There is an increasing attention to the emerging health problem represented by the clinical and functional long-term consequences of SARS-CoV-2 infection, referred to as postacute COVID-19 syndrome. Clinical, radiographic, and autopsy findings have shown that a high rate of fibrosis and restriction of lung function are present in patients who have recovered from COVID-19. Patients with active TB, or those who have recovered from it, have fibrotic scarred lungs and, consequently, some degree of impaired respiratory function. Helminth infections trigger predominantly type 2 immune responses and the release of regulatory and fibrogenic cytokines, such as TGF-β. Here, we analyze the possible consequences of the overlapping of pulmonary fibrosis secondary to COVID-19 and tuberculosis in the setting of sub-Saharan Africa, the region of the world with the highest prevalence of helminth infection.

## 1. Introduction

Reports on the development of fibrotic lesions secondary to coronavirus infection are not new. Clinical, chest computed tomography (CT), and postmortem findings of pulmonary fibrosis (PF) were observed in people who suffered from severe acute respiratory syndrome (SARS) and Middle East respiratory syndrome (MERS), the previous two coronavirus pandemics in the current century [1]. However, the high lethality and the short duration of these pandemics did not allow comprehensive studies to be performed on patients who survived the acute forms of these viral infections.

Even at the beginning of the current pandemic, Paolo Spagnolo et al. [2] predicted that in COVID-19 convalescents, despite the removal of the cause for lung damage, the possibility for the development of progressive and irreversible PF, especially for those with pre-existing pulmonary conditions, may be real. Two years later, the overlapping of PF secondary to COVID-19 with the fibrotic sequelae of other diseases was demonstrated [1,3,4,5]. Here, we analyze the possible consequences of the overlapping of PF secondary to COVID-19 and tuberculosis (TB) in the setting of sub-Saharan Africa (SSA), the region of the world with the highest prevalence of helminth infection.

## 2. Pulmonary Fibrosis (PF) of Postacute COVID-19 Syndrome (PACS)

The natural evolution of SARS-CoV-2 infection can be asymptomatic, evolve with mild symptoms or progress to severe clinical forms. This wide spectrum is a result of triggering host immune responses which, in children and healthy adults, generally contain viral replication at the higher portions of the respiratory system and lead to recovery and, in elderly and patients with comorbidities, can generate an intense pulmonary inflammatory reaction, additional clinical complications, and death [6]. Due to the severe clinical forms of the acute phase of COVID-19, 5,993,901 people had died worldwide as of 8 March 2022 [7].

Although the development and implementation of more effective tools to reduce the incidence and the severity of COVID-19 continues to be a priority, there is an increasing attention to the emerging health problem represented by the unfavorable long-term consequences of that infectious disease. Those adverse consequences include a myriad of clinical manifestations corresponding to injuries in almost all organs, referred to as postacute COVID-19 syndrome (PACS) [8,9].

One important manifestation of PACS is PF, which is a long-lasting and progressive lung disorder caused by excessive deposition of collagen and other extracellular matrix (ECM) components in the organ parenchyma. PF, left to its natural course, can severely impair respiratory function and lead to the development of fatal disability [10,11]. Among the main morphological features of that disorder are the following: (i) an incorrect reestablishment of the injured alveolar epithelium, (ii) fibroblast persistence, (iii) a disproportionate accumulation of ECM components such as collagen, and (iv) the disappearance of regular pulmonary structure [12].

Clinical, chest CT, and postmortem findings have shown that fibrosis and restriction of pulmonary function are frequently present in patients who have recovered from COVID-19: (i) more than a third of the patients with severe forms of COVID-19 may show limited lung function after hospital discharge [12,13], (ii) PF on chest CT have been reported in patients who had recovered from severe or critical disease [12,13,14], and (iii) in an autopsy study involving patients with acute respiratory distress syndrome (ARDS), the finding of PF was more frequent as more time had elapsed since the onset of clinical manifestations of COVID-19 [15]. PF is already recognized among the most important sequelae of SARS-CoV-2 infection [16].

The processes involved in the progress to PF in persons who suffered from acute COVID-19 are complex and, in general, not clearly understood. Several mechanisms, some of them interconnected, have been suggested to explain its development. Of those, three are noteworthy: (i) the downregulation (endocytosis upon virus binding) of the angiotensin-converting enzyme 2 (ACE2) reduces the anti-inflammatory and antifibrotic components of the renin–angiotensin system, leading first to more inflammation and afterwards to fibrosis [17]; (ii) the involvement of type 2 immune cytokines, mainly interleukin-4 (IL-4) and IL-13, each one exhibiting profibrotic activity by promoting the recruitment, activation, and proliferation of the corresponding cellular types [1,10]; and (iii) the increased secretion of transforming growth factor-β (TGF-β), which is a characteristic event in lung fibrotic process [18]. SARS-CoV-2 upregulates TGF-β expression and both TGF-β mRNA as TGF-β protein levels in fibrotic pulmonary tissue are increased [19]. TGF-β can trigger PF by inducing myofibroblast expansion, a key effector event in lung fibrogenesis, and promoting the production and deposition of ECM proteins [20,21].

## 3. Overlapping of Pulmonary Fibrosis (PF) of Postacute COVID-19 Syndrome (PACS) and Tuberculosis (TB)

TB is a chronic and debilitating disease caused by organisms of the *Mycobacterium tuberculosis* (Mtb) complex [22,23]. Although Mtb is primarily a lung pathogen, it can affect practically any organ or tissue. During the last decades, the estimated global TB incidence rate has decreased; nevertheless, TB continues to be an important sanitary problem, mainly in low- and middle-income countries of Africa and Asia, where factors such as poverty, HIV infection, and multidrug resistant TB are fueling the pandemic [23,24].

Depending on the host immune competence, the Mtb infection can evolve from containment, in which the bacteria are isolated within granulomas in latent TB infection (LTBI), to a contagious state, in which the patient will show symptoms that can include cough, fever, night sweats, and weight loss, among others (active TB) [22]. Protection against Mtb requires a distinctly defined type 1 response, mediated by interferon-gamma (IFNγ), IL-2, and tumor necrosis factor-alpha (TNFα), which may clear the infection or restrain it into an immune-mediated containment, also known as latency [23,25]. Active TB is characterized by unlimited mycobacterial multiplication, considerable collagen deposition, and fibrosis [26]. Even though tissue repair during fibrosis is a healing process, large fibrosis with scar formation damages pulmonary function [27].

Early in the pandemic, the World Health Organization (WHO) predicted that patients coinfected with both TB and COVID-19 may have unfavorable clinical evolution [28]. While some studies have not found a significant association of coinfection and disease severity [29,30], others have described a notable higher frequency of undesirable clinical progression among patients with TB and COVID-19 coinfection [31,32].

A recent systematic review and meta-analysis of previous data on the association of COVID-19 and active TB included studies performed almost exclusively in high-TB-burden countries [3]. The overall pooled incidence and lethality found were 1.07% (43 studies) and 1.5% (17 studies), respectively. In agreement with the prediction of WHO, COVID-19 patients with TB had a higher risk of severity (relative risk/risk ratio (RR) 1.46, 95% confidence interval (CI) 1.05–2.02); and mortality (RR 1.93, 95% CI 1.56–2.39), from 20 and 17 studies, respectively, compared to COVID-19 patients without TB.

More recently, another systematic review and meta-analysis performed on data obtained exclusively from high-TB-burden countries of SSA showed that the overall incidence and lethality due to COVID-19/TB coinfection were 2% (20 studies) and 10% (9 studies), respectively [4]. Although the data included in this review corresponded both to people with previous TB and to patients with active infection, the incidence and case-fatality rates of the association were higher than those previously reported [3].

As mentioned above, active TB patients, or those who have recovered from it, are left with fibrotic scarred lungs and, consequently, with some degree of impaired respiratory function. It has been estimated that more than half of all TB survivors have some form of persistent pulmonary dysfunction despite microbiological cure, leaving patients potentially more susceptible to other infectious diseases, included COVID-19 [5]. On the other hand, post-COVID fibrosis may also exacerbate the fibrotic sequelae of pulmonary TB causing a more profound and prolonged disability [3].

## 4. Pulmonary Fibrosis (PF) in the Helminth Coinfection Setting in Sub-Saharan Africa (SSA)

Africa, in particular SSA region, is the continent with the highest prevalence of helminth infections [33]. It is estimated that more than half of SSA’s population is affected by one or more helminth infections, especially by soil-transmitted helminths and schistosomes [23]. Millions of years of host–helminth coevolution have resulted in the development of defensive responses by the hosts and sophisticated immune regulatory mechanisms by the helminths. For more complexity, the immune responses against those parasites, which are relatively large and multicellular organisms, include injury repair processes, necessary to lessen the tissue damage that those pathogens may cause as they move through host organs.

To control helminth infection, the human host typically develops type 2 immune responses (increase of Th2 cells and release of cytokines, primarily IL-4, IL-5, and IL-13) [34,35]. To persist in their host, helminths induce an important immunomodulatory, anti-inflammatory, and fibrogenic pathway: the expansion of FOXP3+ T regulatory cells, B regulatory cells, and M2 macrophages, which together cause the secretion of regulatory cytokines, mainly TGF-β [36]. In relation with this regulatory scenery, it has recently been demonstrated that helminth extracellular vesicles (small membrane-bound vesicles secreted by helminths which contain functional proteins, carbohydrates, lipids, mRNA, and noncoding RNAs) can trigger several events that modulate host–parasite interactions [37,38].

The expanded population of T regulatory cells resulting from the host–helminth interaction can downmodulate both Th1 and Th2 inflammatory responses and interfere with other effector T-cell functions [20,34,35,36]. A prolonged exposure to parasitic helminth infection has been associated with generalized immune hyporesponsiveness [35]. Th2, Tregs, and the immunoregulatory cytokines they produce (such IL-4, IL-5, IL-13, IL-10, and TGF-β) may act as potent inhibitors of the Th1 responses which are required, as it was commented above, for immunity against Mtb infection [23]. Interestingly, some of those cytokines, mainly TGF-β, can trigger PF by promoting the production and deposition of ECM proteins [20,21]. It may be a way for the exacerbation of fibrotic sequelae of pulmonary TB in helminth infection settings.

Despite the underdeveloped economies and limited health care infrastructures of the majority of SSA nations, the lethality of COVID-19 in that region apparently was lower than in the rest of the world during the first part of the pandemic (the time before the massive administration of COVID-19 vaccines in Europe and the Unites States of America).

Some reasons, or combinations of them, have been alluded to explain that unexpected evolution: nonextensive diagnostic testing, age and genetic background of the population, under-reporting of COVID-19 mortality, mutational variations of SARS-CoV-2, environmental temperature and humidity nonfavorable for viral replication, Bacillus Calmette–Guérin (BCG) vaccination policies, composition of the microbiome, endemicity of other infections, and anti-inflammatory component of the helminth immune modulation, among others [39,40,41,42,43].

At times, the prevalence of helminth infections in SSA has been very high [33]. For surviving, helminths modulate the immune responses of their hosts [34,35,36]. The helminths’ immune modulation is highly anti-inflammatory, to the point that allergic and autoimmune events in SSA are relatively rare [35]. The COVID-19 lethality is mainly due to inflammatory phenomena [44]. Some authors have related the relatively low lethality of COVID-19 in SSA with the modulation of immune responses by helminths [41,42,45,46,47,48,49]. Woldey et al. demonstrated that parasite coinfection was associated with a reduced risk of severe COVID-19 in African patients, supporting the hypothesis that parasite coinfection may silence the hyperinflammation associated with severe COVID-19 [49] (Figure 1).

Nevertheless, the long-term outcomes of COVID-19 in SSA may not be as benevolent as the acute phase of the disease as observed in that region. The adverse consequences of the overlapping of fibrotic sequelae of COVID-19 and TB may be amplified by the cytokine profile elicited by helminth coinfection in that setting. As commented above, data obtained from SSA, where the prevalence of helminth infection is very high, showed that the overall incidence and lethality due to COVID-19/TB coinfection were higher there than those reported in other parts of the world [3,4]. The overlapping of post-COVID fibrosis and the fibrotic sequelae of pulmonary TB in a setting of helminth immune modulation (with a predominant type 2 immunity and an increased release of regulatory and fibrogenic cytokines, such as TGF-β), as observed in SSA, may result in more fibrosis and, consequently, a greater disruption of the organ’s architecture (Figure 1).

## 5. Conclusions

Clinical, chest CT, and postmortem findings have shown that a high rate of fibrosis and restriction of lung function are present in patients recovered from COVID-19. Patients with active TB, or those who have recovered from it, have fibrotic scarred lungs and, consequently, some degree of impaired respiratory function. Helminth infections trigger predominantly type 2 immune responses and the release of regulatory and fibrogenic cytokines, such as TGF-β. The overlapping of post-COVID fibrosis and fibrotic sequelae of pulmonary TB, and its adverse clinical consequences, may be amplified by the cytokine profile elicited by helminth coinfection in SSA, the region of the world with the highest prevalence of helminth infection.

Recent findings indicate that the risks of COVID-19 associated with previous and/or current TB may be underestimated in SSA, as this coinfection is under-reported due to logistical constraints [4]. Nevertheless, in spite of the scarcity of accurate data about that association, its fatality rate has been estimated as high [3,4]. For that reason, professionals dealing with TB or COVID-19 patients, mainly in high-burden TB regions, should take into consideration the potential adverse consequences of the association of the fibrotic sequelae of those diseases. The convergence of lung disease after TB and lung disease after COVID-19 necessitates the follow-up of patients with post-TB lung disease who had COVID-19 pneumonia and the prioritization of their linkage to respiratory services for optimal care [50].

At the epidemiological level, the long-term consequences of PF secondary to COVID-19 and TB and their potential amplification by helminth coinfection require more research. Therefore, prospective studies on COVID-19 survivors of SSA populations are necessary to institute better strategies to reduce further disabilities and death in that impoverished region, and possibly in other settings [50].

The administration of effective COVID-19 and TB vaccines may help to decrease the potential burden associated with the COVID-19 and TB PF overlapping in SSA. However, in the development and distribution of vaccines against both pathogens it must be taken into account that helminth infections can impair human immune responses to immuno-gens prepared to control other infectious diseases [51]. Thus, as long as time and resources allow, clinical trials of COVID-19 and TB vaccines to be used in SSA must include the corresponding helminth-infected groups.

In the short-term, and also taking into consideration the global necessity for reducing the number of persons at risk, efforts should be made to speed up the administration of appropriate COVID-19 vaccines in SSA, where, as of March 2022, only 15% of its populations had received a complete schedule of immunization against SARS-CoV-2 infection [7].

## Figures and Tables

**Figure 1 tropicalmed-07-00157-f001:**
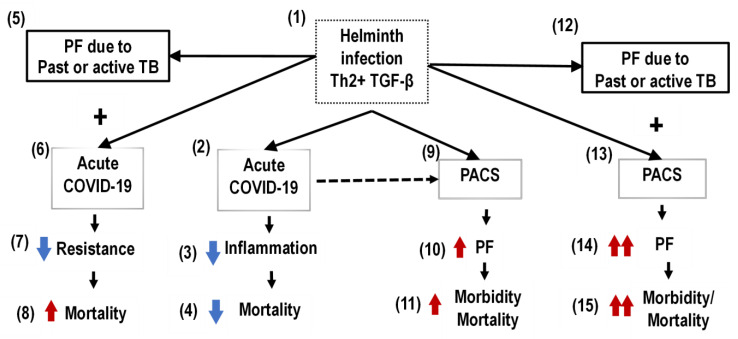
Overlapping of acute COVID-19 and postacute COVID-19 syndrome (PACS) and tuberculosis (TB) in the helminth coinfection setting in sub-Saharan Africa (SSA) population. (1) Helminth coinfection inhibits inflammation and amplifies pulmonary fibrosis (PF) processes; (2–4) helminth coinfection inhibits COVID-19 lung inflammation and decreases mortality; (5–8) infection by *Mycobacterium tuberculosis* and PF due to past or active TB limit resistance to COVID-19 and increase mortality; (9–11) PACS, amplified by helminth infection, increases morbidity and mortality; (12–15) the overlapping of PF due to past or active TB with PF due to PACS, amplified by helminth infection, increases morbidity and mortality.

## Data Availability

Not applicable.

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
