# Peer review of "Overlapping of Pulmonary Fibrosis of Postacute COVID-19 Syndrome and Tuberculosis in the Helminth Coinfection Setting in Sub-Saharan Africa"

_tropicalmed, 2022, doi:10.3390/tropicalmed7080157_

Round 1

Reviewer 1 Report

The manuscript from Luis Fonts et al. brought a critical perspective on considering the COVID-19 related lung sequelae, authors think that the helminth infection created type 2 immune milieus, especially TGF-b generated by the immune regulatory cells, may promote lung fibrosis post SARS-COV2 infection and/or TB infection. This perspective does bring a critical scientific and healthy question for the public health area of SSA. And this perspective should be empathized and followed in the future.

Major:

1.       Please add the references for line 52: Two years later, overlapping of PF secondary to COVID-19 with fibrotic sequelae of other diseases have been demonstrated

2.       Even though due to the limitation of the case studies, the relationship between helminth infection and PF of long COVID is hard to conclude, the TB and helminth coinfection is well established, could the authors discuss more?

Minor:

1.       Line 55-57 look like the writing instruction, it is not supposed to appear in the manuscript.

Author Response

Answer to referee 1

Thank you very much for your revision, comments and suggestions. Please, find below the answers to your queries. As you will note, your opinions and suggestions were taken into account to build the new version of our manuscript.

Major:

  1. Please add the references for line 52: Two years later, overlapping of PF secondary to COVID-19 with fibrotic sequelae of other diseases have been demonstrated.

R/ We agree with the reviewer that references are needed for line 52 (four references are added in the new version of our manuscript).

  1. Even though due to the limitation of the case studies, the relationship between helminth infection and PF of long COVID is hard to conclude, the TB and helminth coinfection is well established, could the authors discuss more?

R/ Thanks for your advice. In the new version of our manuscript we have added a new paragraph to deep into the tuberculosis and helminthic coinfection, on which information is more abundant.

Minor:

  1. Line 55-57 look like the writing instruction, it is not supposed to appear in the manuscript.

R/ We agree with the reviewer that the two sentences between line 55 and 57 look like the writing instructions. For some reason we do not know, that fragment, part of the instructions to the author, appears there. In the new version of our manuscript, that fragment will be removed.

Reviewer 2 Report

In their perspective, Fonte et al. discuss the potential interactions, and some of the possible mechanisms, between pulmonary fibrosis secondary to TB, helminth infections and post-COVID-19 pulmonary fibrosis with a focus on sub-Saharan Africa. While the paper draws attention to the possible implications of the dual burdens of post-COVID-19 pulmonary fibrosis and post-TB pulmonary fibrosis, both possibly influenced by helminths through various pathways, the paper is unable to provide any meaningful conclusions other than that further research is required.

A major concern with the perspective is that it is unclear whether COVID-19 mortality was actually lower sub-Saharan Africa, as the authors assert (Lines 168-171 and pathways 3 and 4 in figure 1). Limited SARS-CoV-2 testing capacity, reporting challenges and absence of civil registration and vital statistics in SSA countries with the notable exception of South Africa substantially limits the validity of almost all analyses of COVID-19 mortality in SSA. Bradshaw’s et al. (https://doi.org/10.17159/sajs.2022/13300) analysis of the excess deaths from South Africa demonstrates that COVID-19 mortality in South Africa was 3 fold higher than officially reported. Furthermore, given the very strong association between age and COVID-19 severity and mortality, age-standardisation is required before any meaningful comparison between countries can be made.

In this regard, the perspective would be substantially strengthened by including more substantial review of the literature on helminth infections and COVID-19 infection and severity and acknowledging the uncertainty in this field. For examples see papers by Wolday https://doi.org/10.1016/j.eclinm.2021.101054 , Abdoli https://dx.doi.org/10.1021/acsptsci.0c00141 , and the letter by Makram https://doi.org/10.1016/j.ijid.2022.03.027.

Author Response

Answer to referee 2

Thank you very much for your revision, comments and suggestions. Please, find below the answers to your queries. As you will note, your opinions and suggestions were taken into account to build the new version of our manuscript.

1- “In their perspective, Fonte et al. discuss the potential interactions, and some of the possible mechanisms, between pulmonary fibrosis secondary to TB, helminth infections and post-COVID-19 pulmonary fibrosis with a focus on sub-Saharan Africa. While the paper draws attention to the possible implications of the dual burdens of post-COVID-19 pulmonary fibrosis and post-TB pulmonary fibrosis, both possibly influenced by helminths through various pathways, the paper is unable to provide any meaningful conclusions other than that further research is required.”

R/ Thanks for this opportune comment. In the new version of our manuscript we have expanded our conclusions.

2- “A major concern with the perspective is that it is unclear whether COVID-19 mortality was actually lower sub-Saharan Africa, as the authors assert (Lines 168-171 and pathways 3 and 4 in figure 1). Limited SARS-CoV-2 testing capacity, reporting challenges and absence of civil registration and vital statistics in SSA countries with the notable exception of South Africa substantially limits the validity of almost all analyses of COVID-19 mortality in SSA. Bradshaw’s et al. (https://doi.org/10.17159/sajs.2022/13300) analysis of the excess deaths from South Africa demonstrates that COVID-19 mortality in South Africa was 3 fold higher than officially reported. Furthermore, given the very strong association between age and COVID-19 severity and mortality, age-standardisation is required before any meaningful comparison between countries can be made”.

R/ Thank you for your very interesting comment on a controversial topic. As controversial as it has motivated the publication of not a few Opinion or Perspective articles, as our manuscript tries to be. It is very difficult to refer to your exciting comment in a few words, I will try.

We, like other authors who have tried to explain it in their respective papers, consider that, despite the under-developed economies and limited health care infrastructures of the majority of SSA nations, the lethality (not incidence) of COVID-19 in that region was lower than in the rest of the world during the first part of the pandemic (the time before the massive administration of COVID-19 vaccines in Europe and Unites States of America). We read the Bradshaw’s et al. paper. Those colleagues did not demonstrate that COVID-19 mortality in South Africa was 3 fold higher than officially reported. Bradshaw’s et al. wrote “From this we see that reported COVID-19 deaths account for around one third of the excess natural deaths…”, “…we are unable to ascertain with certainty the proportion of excess deaths attributable to COVID-19 from the South African death data, …”, and below the heading Study limitations the authors wrote “In addition, there is considerable uncertainty around what proportion of the excess deaths was due to COVID-19 (directly or indirectly) and the true range of uncertainty about the estimate of excess deaths”. And this is very important, we must bear in mind that COVID-19 caused many deaths not directly related to the virus, many health systems were overwhelmed by the pandemic and control programs for other diseases, including non-infectious ones, were neglected. Finally, we agree with Bradshaw’s et al. when they wrote “However, South Africa is not a bellwether for Africa: its population is somewhat older than that of most African countries and has higher prevalence of co-morbidities, being relatively wealthier than most other African countries. In addition, South Africa has a very high HIV burden”. The epidemiological characteristics of the rest of SSA are not those of South Africa. For example, the rest of SSA is located further north, in climatologically more tropical areas, where other infectious agents, including helminths, are more prevalent.

Waogodo et al. in a recent article published in Lancet Glob Health (Lancet Glob Health 2022 https://doi.org/10.1016/S2214-109X(22)00233-9) addresses the issue of deaths in SSA by COVID-19 using a different analysis model and reaches the following conclusions: “With the deaths, however, the reported numbers are closer to the true burden, which is a reflection of the relative difficulty in missing mortality statistics compared with missing infections, many of which present with non-troubling symptoms. The estimated deaths represent an infection fatality rate of 0·09% (0·03% for 2020 and 0·16% for 2021), which is comparable to the value of 0·09% predicted for locations with COVID-19 population mortality rates less than the global average (<118 deaths per million)”, and “In summary, the findings of our model suggest that the WHO African region is estimated to have had a similar number of COVID-19 infections to the rest of the world, but with fewer deaths”.

Dear colleague, our manuscript is not related to the deaths of the acute phase of COVID-19 in SSA. Our objective is to analyze the possible consequences of the overlapping of pulmonary fibrosis secondary to COVID-19 and tuberculosis in the setting of Sub-Saharan Africa, the region of the world with the highest prevalence of helminth infection. Between lines 168 and 173 of the version you reviewed we briefly referred to the acute phase of COVID-19 in SSA. TAKEN INTO ACCOUNT YOUR COMMENT, IN THE NEW VERSION OF OUR MANUSCRIPT WE WILL MENTION OTHER FACTORS THAT MAY BE RELATED WITH THE LOWER LETHALITY OF COVID-19 IN SSA. THANKS AGAIN.

3-“In this regard, the perspective would be substantially strengthened by including more substantial review of the literature on helminth infections and COVID-19 infection and severity and acknowledging the uncertainty in this field. For examples see papers by Wolday https://doi.org/10.1016/j.eclinm.2021.101054 , Abdoli https://dx.doi.org/10.1021/acsptsci.0c00141, and the letter by Makram https://doi.org/10.1016/j.ijid.2022.03.027”.  

R/ For times, the prevalence of helminth infections in SSA has been very high. For surviving, helminths modulate the immune responses of their hosts. The modulation of immune responses by helminths is highly anti-inflammatory, to the point that allergic and autoimmune events in SSA are relatively rare. The COVID-19 lethality is mainly due to inflammatory phenomena. In the fragment of our manuscript you refer (Lines 168-171 and pathways 3 and 4 in figure 1), as in another article we published on the subject (Fonte et al. Frontiers in immunology 2020, https:// doi: 10.3389/fimmu.2020.574910) we relate the helminth immune modulation with the lower COVID-19 lethality in SSA, NOT WITH INCIDENCE OF THIS VIROSIS THERE. The letter of Makram, that you suggest us, briefly analyzes opinions on the subject. The paper of Wolday et al., that you suggest us, supports our opinion. The article of Abdoli et al., that you suggest us, not diverges of our opinion at all. Abdoli et al. summarize “Helminth infections are among the most common infectious diseases in underdeveloped countries. Helminths suppress the host immune responses and consequently mitigate vaccine efficacy and increase severity of other infectious diseases. Helminth co-infections might suppress the efficient immune response against SARS-CoV-2 at the early stage of the infection and may increase morbidity and mortality of COVID-19”. We differ partially of Abdoli et al. Like them, we consider “The consequence of this immune modulation is suppression of the essential immune response against intracellular pathogens. Therefore, in individuals with helminth infections, the susceptibility and severity to infectious diseases, such as HIV/AIDS, tuberculosis, and malaria, have increased. As reviewed elsewhere, helminth infections are more common among developing or underdeveloped regions of the world, such as sub-Saharan Africa, East Asia, and the Indian subcontinent. In these regions, the prevalence of major infectious diseases, including HIV/AIDS, malaria, and tuberculosis, are higher than that in other parts of the world”. TAKEN INTO ACCOUNT THAT SARS COV-2 IS AN INTRACELLULAR MICROORGANISM, LIKE ABDOLI ET AL., WE CONSIDER THAT THE INCIDENCE OF COVID-19 MAY BE INCREASED IN REGION WHERE HELMINTHS ARE PREVALENT. However, the modulation of immune responses by helminths is highly anti-inflammatory and the COVID-19 lethality is mainly due to inflammatory phenomena. THUS, IN AGREEMENT WITH WOLDAY ET AL., WE BELIEVE THAT THE LETHALITY OF COVID-19 IN SSA COULD BE REDUCED, AMONG OTHER FACTORS, BY THE MODULATION OF IMMUNE RESPONSES BY HELMINTHS.

Dear colleague, although our manuscript is not related to the deaths of the acute phase of COVID-19 in SSA (our objective is to analyze the possible consequences of the overlapping of pulmonary fibrosis secondary to COVID-19 and tuberculosis in the setting of Sub- Saharan Africa), TAKEN INTO ACCOUNT YOUR SUGGESTION, IN THE NEW VERSION OF OUR MANUSCRIPT WE WILL INCLUDE A BRIEF COMMENT TO THE POSSIBLE INFLUENCE OF HELMINTH IMMUNE MODULATION ON COVID-19 LETHALITY. THANKS FOR YOUR SUGGESTION THAT COULD PERMIT A MORE HOLISTIC COMPREHENSION OF OUR MANUSCRIPT,

Round 2

Reviewer 2 Report

Thank you for submitting a revised manuscript. 

I suggest the authors consider inclusion of under-reporting of COVID-19 mortality as one of the reasons for the apparently lower COVID-19 mortality in SSA (lines 186-191). 

Author Response

Dear colleaghe,

In the version of our manuscript we are uploading now, we have included the phrase you suggest.

Thanks again,
